
# Quantum Gross-Pitaevskii Equation

Jutho Haegeman[1], Damian Draxler[2], Vid Stojevic[1,3] J. Ignacio Cirac[4],
Tobias J. Osborne[5] and Frank Verstraete[1,2]

**1** Department of Physics and Astronomy, University of Ghent, Krijgslaan 281 S9,
B-9000 Ghent, Belgium
**2** Faculty of Physics, University of Vienna, Boltzmanngasse 5, A-1090 Wien, Austria
**3** London Centre for Nanotechnology, 17-19 Gordon St, London, WC1H 0AH
**4** Max-Planck-Institut für Quantenoptik, Hans-Kopfermann-Str. 1, Garching,
D-85748, Germany
**5** Institut für Theoretische Physik, Leibniz Universität Hannover, Appelstr. 2,
30167 Hannover, Germany

## Abstract

**We introduce a non-commutative generalization of the Gross-Pitaevskii equation for one-dimensional quantum gasses and quantum liquids. This generalization is obtained by applying the time-dependent variational principle to the variational manifold of continuous matrix product states. This allows for a full quantum description of many body system —including entanglement and correlations— and thus extends significantly beyond the usual mean-field description of the Gross-Pitaevskii equation, which is known to fail for (quasi) one-dimensional systems. By linearizing around a stationary solution, we furthermore derive an associated generalization of the Bogoliubov – de Gennes equations. This framework is applied to compute the steady state response amplitude to a periodic perturbation of the potential.**

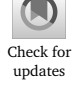

## 1  Introduction

In 1961, Gross and Pitaevskii developed the mean-field theory description of cold Bose gasses [1–3], which resulted in the ubiquitous Gross-Pitaevskii equation (GPE)

$$\mathrm{i}\frac{\partial \phi(\boldsymbol{x},t)}{\partial t} = (-\Delta + v(\boldsymbol{x}))\phi(\boldsymbol{x},t) + 2g|\phi(\boldsymbol{x},t)|^2\phi(\boldsymbol{x},t), \tag{1}$$

where $\phi(x,t)$ is the order parameter of the Bose-Einstein condensate in a trapping potential $v(x)$. Ever since, this equation constitutes the cornerstone for the theoretical description of cold atomic gasses [4–6]. Its success can be explained by the fact that the GPE agrees with the full quantum solution for the three-dimensional problem in the weak-density limit, which corresponds with the typical experimental setup of trapped dilute gasses (see Ref. [7] and references therein). By linearizing around stationary solutions of the GPE, one obtains the Bogoliubov – de Gennes equations (BdGE), which describe small scale excitations on top of the background state and can be used to compute linear response to perturbations.

The GPE without trapping potential $[v(x) = 0]$ is also known as the nonlinear Schrödinger equation and appears in several areas of theoretical physics as it offers a canonical description for slowly varying, quasi-monochromatic wave packets in dispersive, weakly nonlinear media [8,9]. As such, it has also stimulated an abundance of mathematical research towards showing the stability of its (solitary wave) solutions [10–12], as well as towards the development of numerical integrators [13–17].

In the case of one spatial dimension [18,19], as realized in highly elongated traps [20–24], both the GPE [25] and the full quantum mechanical problem known as the Lieb-Liniger model [26–28] are integrable for constant potential, but the respective solutions do not agree. One-dimensional Bose gasses have no condensation (only quasi long-range order) [29], show quasi-fermionic behavior [30–33] and have excitations which cannot be predicted from Bogoliubov's theory [27, 34]. This behavior has no classical counterpart and is dominated by quantum correlations. This paper develops a generalization of the one-dimensional GPE and the BdGE, where quantum correlations are taken into account. They are formulated in terms of non-commuting matrices and —following the typical nomenclature of integrable systems— are referred to as the *quantum Gross-Pitaevskii equation* (QGPE) and *quantum Bogoliubov – de Gennes equations* (QBdGE). We apply the latter to compute linear response behaviour in the density profile when a periodic perturbation to the potential is applied.

The normal GPE can be derived by applying the Dirac-Frenkel time-dependent variational principle (TDVP) [35–37] to a variational mean field ansatz $|\Psi[\phi]\rangle$ in the canonical $[|\Psi[\phi]\rangle \sim (\int \phi(x)\hat{\psi}^\dagger(x)\,\mathrm{d}x)^N |\Omega\rangle$ for $N$ particles] or grand-canonical $[|\Psi[\phi]\rangle \sim \mathrm{e}^{\int \phi(x)\hat{\psi}^\dagger(x)\,\mathrm{d}x} |\Omega\rangle]$ ensemble, where $\hat{\psi}^\dagger(x)$ is the bosonic field creation operator in second quantization and $|\Omega\rangle$ is the Fock vacuum. For one-dimensional bosonic systems, the GPE is obtained by applying this ansatz to the Lieb Liniger Hamiltonian [26]

$$\hat{H} = \int \mathrm{d}x \, \frac{\mathrm{d}\hat{\psi}^{\dagger}}{\mathrm{d}x}(x) \frac{\mathrm{d}\hat{\psi}}{\mathrm{d}x}(x) + v(x)\hat{\psi}^{\dagger}(x)\hat{\psi}(x) + g\hat{\psi}^{\dagger}(x)\hat{\psi}^{\dagger}(x)\hat{\psi}(x)\hat{\psi}(x). \tag{2}$$

The variational manifold of *continuous matrix product states* (cMPS) [38–40] can be seen as a generalization of this grand-canonical ansatz in which the variational function $\phi(x)$ is replaced by a matrix valued function $R(x)$:

$$|\Psi[R, \boldsymbol{v}_1, \boldsymbol{v}_2]\rangle = \boldsymbol{v}_1^{\dagger} \mathcal{P} \mathrm{e}^{\int_{x_1}^{x_2} R(x) \otimes \hat{\psi}^{\dagger}(x) \, \mathrm{d}x} \boldsymbol{v}_2 \, |\Omega\rangle. \tag{3}$$

Here $\mathcal{P}$ denotes the path-ordered exponential and $\boldsymbol{v}_{1,2}$ are $D$-dimensional boundary vectors. By choosing the bond dimensions $D = 1$, we clearly recover the mean field ansatz. The cMPS ansatz was conceived as a continuum limit of the matrix product state (MPS) ansatz [41–43], which underlies the highly successful density matrix renormalization group [44] for the description of one-dimensional quantum spin systems. By enlarging the refinement parameter $D$, the exact quantum state can be increasingly well approximated. Indeed, the cMPS ansatz was shown to represent both the ground state [38] and the two types of elementary excitations [45] of the Lieb-Liniger model very well for moderate values of $D$. The goal of this paper is to apply the TDVP formalism to the cMPS manifold in order to derive the matrix or quantum analogue of the GPE.

## 2 Quantum Gross-Pitaevskii Equation

For a complex manifold, the TDVP can be understood as a replacement of Schrödinger's equation by

$$\mathrm{i}\frac{\mathrm{d}}{\mathrm{d}t} |\Psi\rangle = \hat{P}_{\Psi}\hat{H} |\Psi\rangle, \tag{4}$$

where $P_{\Psi}$ is a projector onto the tangent space of the variational manifold at the point $|\Psi\rangle$. Whereas the Schrödinger equation would immediately take an initial state away from the variational manifold, this extra projector assures that the evolution remains within the manifold. The QGPE can thus be obtained by finding a time derivative $\partial_t R$ (and $\partial_t \boldsymbol{v}_{1,2}$) such that $\mathrm{d} |\Psi[R, \boldsymbol{v}_1, \boldsymbol{v}_2]\rangle / \mathrm{d}t$ has the same inner product as $\hat{H} |\Psi\rangle$ with any possible tangent vector. Using the expressions of tangent vectors and overlaps with Hamiltonians obtained in [40], we obtain [see Supplementary Material for details]

$$\begin{aligned} \mathrm{i}\partial_t R(x) = \left(-\partial_x^2 + v(x)\right)R(x) + g\left(\rho_L^{-1}(x)R^{\dagger}(x)\rho_L(x)\right)R^2(x) + gR^2(x)\left(\rho_R(x)R^{\dagger}(x)\rho_R^{-1}(x)\right) \\ - \left(\rho_L^{-1}(x)R^{\dagger}(x)\rho_L(x)\right)[R(x), \partial_x R(x)] - [R(x), \partial_x R(x)]\left(\rho_R(x)R^{\dagger}(x)\rho_R^{-1}(x)\right), \end{aligned} \tag{5}$$

where $\rho_L(x)$ and $\rho_R(x)$ are $D \times D$ reduced density matrices defined by the equations

$$\rho_L(x_1) = \boldsymbol{v}_1\boldsymbol{v}_1^{\dagger}, \qquad\qquad \partial_x \rho_L(x) = R^{\dagger}(x)\rho_L(x)R(x), \tag{6a}$$

$$\rho_R(x_2) = \boldsymbol{v}_2\boldsymbol{v}_2^{\dagger}, \qquad\qquad -\partial_x \rho_R(x) = R(x)\rho_R(x)R^{\dagger}(x). \tag{6b}$$

Note that the original GPE can be read off from the first line of (5) for $D = 1$, while the second line —involving a commutator $[R(x), \partial_x R(x)]$— has no mean field analogue. Since the non-vanishing of this term is tantamount to the presence of quantum correlations, it would be extremely interesting to investigate its physical consequences in more detail.

## 2.1 Gauge invariance

As is well known in the literature of MPS, efficient and robust algorithms make crucial use of gauge transforms, i.e. transformations of the kind $R(x) \rightarrow G^{-1}(x)R(x)G(x)$ for an arbitrary matrix function $G(x)$ that leaves the physics invariant. A manifestly gauge invariant QGPE is obtained by introducing two more $D \times D$ matrix valued functions $P(x)$ and $Q(x)$, which can be interpreted as the $A_0$ and $A_1$ components of a gauge potential. The spatial and temporal derivates are then replaced by covariant derivatives

$$\partial_x R(x) \rightarrow \mathcal{D}_x R(x) \equiv \partial_x R(x) + [Q(x), R(x)] \tag{7a}$$

$$\partial_t R(x) \rightarrow \mathcal{D}_t R(x) \equiv \partial_t R(x) + [P(x), R(x)]. \tag{7b}$$

and the cMPS acquires the conventional form [38]

$$|\Psi[Q,R,\boldsymbol{v}_1,\boldsymbol{v}_2]\rangle = \boldsymbol{v}_1^\dagger \mathcal{P} e^{\int_{x_1}^{x_2} Q(x) \otimes \hat{I} + R(x) \otimes \hat{\psi}^\dagger(x) \, dx} \boldsymbol{v}_2 \, |\Omega\rangle .$$

The manifestly covariant QGPE becomes

$$i\mathcal{D}_t R(x) = \left(-\mathcal{D}_x^2 + v(x)\right)R(x) + g\left(\rho_L^{-1}(x)R^\dagger(x)\rho_L(x)\right)R^2(x) + gR^2(x)\left(\rho_R(x)R^\dagger(x)\rho_R^{-1}(x)\right)$$
$$-\left(\rho_L^{-1}(x)R^\dagger(x)\rho_L(x)\right)[R(x), \mathcal{D}_x R(x)] - [R(x), \mathcal{D}_x R(x)]\left(\rho_R(x)R^\dagger(x)\rho_R^{-1}(x)\right) \tag{8}$$

in combination with a new equation for the time evolution of $Q(x)$

$$i\partial_t Q(x) - i\partial_x P(x) - i[Q(x), P(x)] = -\rho_L(x)^{-1}R(x)^\dagger \rho_L(x) \times$$
$$\left\{gR(x)^2 - [R(x), \mathcal{D}_x R(x)]\right\}\rho_R(x)R(x)^\dagger \rho_R(x)^{-1}. \tag{9}$$

Note that the left hand side can be recognised as the only nonzero component $F_{0,1}$ of the antisymmetric field tensor. The defining equations for the density matrices $\rho_{L,R}$ are changed to $\partial_x \rho_L(x) = Q(x)^\dagger \rho_L(x) + \rho_L(x)Q(x) + R^\dagger(x)\rho_L(x)R(x)$ and similarly for $\rho_R(x)$. These equations as well as the corresponding cMPS are invariant under arbitrary $x$- and $t$-dependent gauge transformation $G(x,t) \in \mathsf{GL}(D)$

$$\rho_L(x,t) \rightarrow G^\dagger(x,t)\rho_L(x,t)G(x,t)$$
$$\rho_R(x,t) \rightarrow G^{-1}(x,t)\rho_R(x,t)G^{\dagger-1}(x,t)$$
$$R(x,t) \rightarrow G^{-1}(x,t)R(x,t)G(x,t)$$
$$Q(x,t) \rightarrow G^{-1}(x,t)\left(Q(x,t) + \partial_x\right)G(x,t)$$
$$P(x,t) \rightarrow G^{-1}(x,t)\left(P(x,t) + \partial_t\right)G(x,t). \tag{10}$$

A great benefit from working in this representation is that the matrices $R(x,t), Q(x,t), P(x,t)$ can be chosen to be independent of $x$ for translational invariant systems, greatly reducing the complexity of integrating the QGPE. Furthermore, it allows to fix the cMPS to remain in e.g. the left canonical form $\rho_L(x,t) = \mathbb{1}$ [and thus $Q(x)^\dagger + Q(x) + R(x)^\dagger R(x) = 0$] by choosing $P(x) = -iR^\dagger(x)\mathcal{D}_x R(x) + iF(x)$ with $F(x)$ a hermitian matrix. This $F(x)$ can be chosen freely in the case of real time evolution, but has to properly chosen in the case of imaginary time evolution. One particular choice is the solution of

$$\partial_x F - Q^\dagger F - FQ - R^\dagger FR = (\mathcal{D}_x R)^\dagger(\mathcal{D}_x R) + v(x)R^\dagger R + g(R^\dagger)^2 R^2 , \tag{11}$$

which ensures that $\partial_t Q(x) + R(x)^\dagger \partial_t R(x) = 0$.

## 2.2 Boundary Conditions

For finite systems, the QGPE needs to be supplemented with appropriate boundary conditions to fully specify the problem. These will also affect the evolution equation for the boundary vectors $\mathbf{v}_{1,2}$, which we derive presently. As the TDVP can be obtained from extremizing an action, there are only two types of self-consistent boundary conditions (similar to *e.g.* the classical wave equation for a vibrating string), unless explicit boundary terms are included in the Hamiltonian from Eq. (2). These can be derived by considering the quantized field operator $\hat{\psi}(x)$ and expressing stability with respect to variations of $\hat{\psi}^{\dagger}(x)$. When the value of $\hat{\psi}(x)$ is fixed at the boundaries, Dirichlet conditions are obtained:

$$\hat{\psi}(x_1) = a \qquad \Rightarrow \qquad \mathbf{v}_1^{\dagger} R(x_1) = a \mathbf{v}_1^{\dagger}, \tag{12a}$$

$$\hat{\psi}(x_2) = b \qquad \Rightarrow \qquad R(x_2) \mathbf{v}_2 = b \mathbf{v}_2. \tag{12b}$$

Alternatively homogenous Neumann or mixed boundary conditions could be used. While the resulting boundary conditions for the variational parameters $R$ are inherently gauge invariant, they only correspond to $D$ instead of $D^2$ equations each. In e.g. the case of Dirichlet conditions, they only fix one eigenvalue and eigenvector of the matrix $R$. In fact, the other directions of $R$ at the boundary do not appear in physical expectation values and can thus not be fixed from physical considerations. In order to eliminate any interplay with the gauge transformation, we will 'promote' the boundary conditions in a gauge invariant manner by imposing them as identity matrix, e.g. $R(x_1) = a\mathbb{1}_D$ in the case of Eq. (12a). Note that there are no separate boundary condition on $Q(x)$, as these degrees of freedom can be interpreted as pure gauge degrees of freedom. The boundary conditions then also affect the TDVP equation for the boundary vectors. For the case of Dirichlet conditions $[R(x_1) = a\mathbb{1}_D$ and $R(x_2) = b\mathbb{1}_D]$, we find

$$i\partial_t \mathbf{v}_1^{\dagger} - i\mathbf{v}_1^{\dagger} P(x_1) = -\overline{a}\mathbf{v}_1^{\dagger} \mathcal{D}_x R(x_1) \tag{13a}$$

$$i\partial_t \mathbf{v}_2 + iP(x_2)\mathbf{v}_2 = +\overline{b}\mathcal{D}_x R(x_2)\mathbf{v}_2 \tag{13b}$$

where the left hand side contains the covariant time derivative in the conjugate and fundamental representation, respectively. These equations are also valid for the Neumann conditions, where the right hand side becomes zero.

## 2.3 Symplectic structure

Let us now discuss in more detail the mathematical structure of the QGPE. Since it contains the quantities $\rho_{L,R}(x)$, which are defined by integrating Eq. (6), it forms a set of coupled non-linear partial integro-differential equations[1] containing first order time derivatives and second order space derivatives. It is a non-commutative generalization of the normal GPE in that it is defined in terms of matrix variables. Indeed, the normal GPE is recovered from Eq. (8) in the limit $D = 1$ by setting $R(x) = \phi(x)$ and observing that commutators then vanish. In that limit, $\int_{x_1}^{x_2} Q(x)\,\mathrm{d}x$ acts as on overall scalar factor (norm and phase) that can be absorbed in the boundaries.

Just like the normal GPE and essentially any TDVP equation, the real-time QGPE evolution forms a classical Hamiltonian system where $\langle\Psi[\overline{Q},\overline{R},\overline{\mathbf{v}}_1,\overline{\mathbf{v}}_2]|\hat{H}|\Psi[Q,R,\mathbf{v}_1,\mathbf{v}_2]\rangle$ plays the role of the classical Hamiltonian. The resulting differential equations are therefore symplectic and the energy expectation value is a constant of motion when $\hat{H}$ is time-independent [37][2]. When

---

[1]One can also formulate time evolution differential equations for $\rho_{L,R}(x)$ in order to make it into a larger set of ordinary non-linear partial differential equations.

[2]The symplectic and geometric structure are only compatible for complex submanifolds of Hilbert space, which are automatically Kähler with the Hermitian metric determined by the physical overlap of any two tangent vectors to the manifold

using the QGPE with imaginary time evolution $t \to -i\tau$ to find a cMPS approximation for the quantum ground state, this symplectic structure is of course lost and the energy expectation value decreases monotonically until convergence.

## 2.4 Numerical integration

Developing a stable numerical integration scheme for the QGPE is challenging. Firstly, there are the inherent complexities associated with solving a set of non-linear partial integro-differential equations. Because of the second order spatial derivative and first order time derivative, the Courant-Friedrichs-Levy condition limits the time step of explicit schemes. A typical workaround for the GPE is to use a splitting scheme [13, 15, 17], where the evolution is decomposed into the local terms (external potential and interaction) and the kinetic term. The linearity of the latter allows for a solution using a Crank-Nicolson method [46] or in Fourier space [47]. While the QGPE is still linear in the second order spatial derivative, it has nonlinear terms containing first-order spatial derivatives, which cannot easily be integrated in Fourier space. Another complication of the QGPE, as formulated in Eq. (13), is that it depends on the inverses of the density matrices $\rho_L(x)$ and $\rho_R(x)$. These become rank deficient near respectively the left and right boundary, as is clear from the definition in Eq. (6). It is well known in the tensor network community that the gauge degrees of freedom in the underlying matrix product state have to be exploited to transform these density matrices into identity matrices [48]. This was implemented only recently for the TDVP equation for matrix product states [49], using a non-trivial decomposition of the tangent space projector that allows to split the non-linear differential equation for all variables into a set of linear differential equations for the individual MPS tensors [50], which is made possible by the fact that the MPS parameterization is multilinear. Here too, we have to face complications introduced by the intrinsic nonlinearity in the cMPS parameterization. A final challenge is to develop suitable continuum analogous of the factorization routines such as the QR- or singular value decomposition, which are exploited in the MPS simulations to robustly implement the required gauge transformations. Note, in addition, that in order to exploit gauge freedom, the discretization of the QGPE should not break gauge invariance. Hereto, ideas from lattice gauge theory can serve as inspiration. In the translation invariant setting (when $Q$ and $R$ become $x$-independent matrices), several of these difficulties disappear.

## 3 Quantum Bogoliubov-de Gennes equations

In the case of small perturbations around a translational invariant Hamiltonian, it might therefore be useful to linearize the QGPE around the translation invariant cMPS. The ensuing equations are "quantum" versions of the Bogoliubov-de Gennes equations. As an example, let us assume that we have found a variational minimum $R_0, Q_0$ and corresponding $\rho_{L0} = \hat{I}$ and $\rho_{R0}$ for the ground state of a translation invariant Hamiltonian $H_0$. We now wish to study the response of the system when applying an external potential $\epsilon V(x, t)$. We can readily expand the arguments of the QGPE (13) to first order $R(x, t) = R_0 + \epsilon \tilde{R}(x, t)$, $Q(x, t) = Q_0 + \epsilon \tilde{R}(x, t)$, $\rho_R(x, t) = \rho_{R0} + \epsilon \tilde{\rho}_R(x, t)$ and obtain linear equations for all new variables. As the QGPE mixes $R$ with its conjugate, the corresponding equations decouple all different Fourier modes from each other except those with opposite momenta, leading to a simple linear set of equations with $2D^2$ unknowns. In particular, if we consider a time-dependent perturbation of the form $\hat{H}_1 = \int dx v(t) \cos(kx - \omega t) \psi^\dagger(x) \psi(x)$, the equation for $\tilde{R} = e^{i(kx-\omega t)} R_+ + e^{-i(kx-\omega t)} R_-$

becomes [3]

$$\begin{bmatrix} \omega & 0 \\ 0 & -\omega \end{bmatrix} \begin{bmatrix} R_+ \\ R_-^\dagger \end{bmatrix} = \begin{bmatrix} H_{\text{eff}} & M_{\text{eff}} \\ M_{\text{eff}}^\dagger & H_{\text{eff}} \end{bmatrix} \begin{bmatrix} R_+ \\ R_-^\dagger \end{bmatrix} + \begin{bmatrix} v_1 \\ v_2 \end{bmatrix}. \tag{14}$$

The matrix appearing on the right hand side of the above expression is (the momentum $\pm k$ block of) the Hessian of the energy functional $E(\overline{R}_0, \overline{Q}_0, R_0, Q_0) = \langle \Psi[\overline{Q}_0, \overline{R}_0] | \hat{H}_0 | \Psi[Q_0, R_0] \rangle$, whereas the inhomogeneous vector $\begin{bmatrix} v_1 & v_2 \end{bmatrix}$ contains the driving terms from $\hat{H}_1$. Note that Eq. (14) reduces to a normal eigenvalue problem when $\hat{H}_1 = 0$ and is related to the ansatz for excitations introduced in Ref. [45], which is capable of faithfully reproducing Type II Lieb-Liniger excitations.

As an application, let us use this formalism to compute the change in particle density $\delta \langle \rho(x) \rangle = \langle \rho(x) \rangle - \rho$ on top of translation invariant Lieb Liniger solutions with constant density $\rho$ for a small static ($\omega = 0$) perturbation with varying wave vector $k$, for different values of the interaction strength $g$. Because we have linearized the QGPE in order to arrive at the above equation, the density fluctuation $\delta \langle \rho(x) \rangle$ will be directly proportional to the strength $v$ of the perturbation in the potential, which we set equal to unit value for convenience. We study the linear response amplitude as function of the interaction strength $\gamma = g/\rho$ and as a function of the wave vector $k$ of the perturbation. This response should be observable in experiments akin to those of Refs. [51–53]. Our simulation results are presented in Fig. 1.

The Bogoliubov mean field result is given by

$$\langle \delta \rho(x) \rangle = -\frac{2\rho k^2}{k^4 + 4\gamma \rho^2 k^2} v \cos(kx), \tag{15}$$

with the fraction representing the mean field result for the response amplitude $\alpha$ shown in the top panel of Fig. 1. It only matches our solution for small values of $\gamma$. For larger values of $\gamma$ (stronger interactions), our results indicate a strong response around $k = 2k_F$, with $k_F = \pi\rho$ the Fermi momentum. This can be well understood from the Tonks-Girardeau limit $\gamma \to \infty$ [30]. This response peak is thus a clear signature of the effect of Lieb's Type II excitations [27], which effectively arise on top of the strongly correlated ground state induced by the interactions and cannot be captured by Bogoliubov's theory. In contrast, the mean field result is dominated by the Type I excitations. Indeed, the mean field dispersion relation appears in the denominator of the fraction in Eq. (15). In summary, our framework provides results which are consistent throughout the whole range of interaction strengths.

## 4  Conclusion and outlook

We have developed a natural generalization of the GPE based on the formalism of cMPS suited for the study of one-dimensional quantum systems where entanglement plays an important role and the mean-field ansatz underlying the GPE is not justified. While it would be interesting to have a complete mathematical study of the existence, uniqueness (up to gauge transformations) and stability of the solutions of the QGPE for given boundary conditions, this is beyond the goal and scope of this paper. It would also be enlightening to investigate whether there exist quantum generalization of certain exact solutions of the GPE, such as the dark soliton solution, and how they relate to the integrability of the full Lieb-Liniger Hamiltonian or to the topological excitations constructed using the related ansatz of Ref. [45]. Note that integrable matrix versions of the GPE were already formulated for the mean-field description of multicomponent Bose-Einstein condensates [54–56]. In addition, it would be instructive to compare the predictions of the QGPE to existing beyond-mean field studies such as e.g.

---

[3]See Supplementary Material for further details.

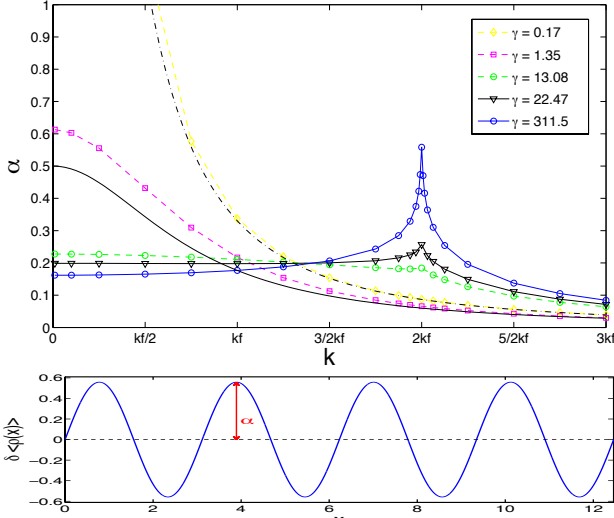

Figure 1: Top: The response amplitude $\alpha = \max_x \delta \langle \rho(x) \rangle$ for different values of $\gamma = g/\rho$ as a function of the momentum $k$. The dashed and solid line without markers are the classical Bogoliubov results for $\gamma = 0.17$ and $\gamma = 1.35$ respectively. Bottom: Density fluctuation $\delta \langle \rho(x) \rangle = \langle \rho(x) \rangle - \rho$ at $\gamma = 311.5$ and $k = 2k_{\mathrm{F}}$. All calculations in both panels were done with cMPS bond dimension $D = 64$.

Ref. [57]. Like the phase space methods here proposed, the QGPE might similarly be restricted to short simulation times for dynamical problems. The underlying physical reason is however completely different and caused by the growth of entanglement in such settings, which cannot be captured by the underlying variational cMPS ansatz.

As the cMPS ansatz is a versatile variational ansatz that can readily be extended to bosonic and fermionic systems with *e.g.* multiple particle species [40, 58–60], the QGPE equations generalize straightforwardly to such systems. Moreover, the approach described in this paper is in no way restricted to the Lieb-Liniger Hamiltonian, but is applicable to arbitrary Hamiltonians in one spatial dimension. cMPS methods have already been used to study (1+1) dimensional relativistic theories for fermions [58] and bosons [61] in a translationally invariant setting. Using the regularisation method described in Ref. [61], the derivation of the QGPE presented in this paper extends straightforwardly to such systems, enabling in principle the study of general bosonic non-linear $\sigma$-models with boundaries. Non-linear $\sigma$-models are of significant interest for the high energy physics community, for example, providing the underlying description of bosonic strings [62, 63] propagating on curved backgrounds. Given that cMPS methods are intrinsically non-perturbative, the approach of this paper has the potential to be particularly useful in the study of such models in the limit of large curvature, when perturbative quantization schemes fail. It is furthermore very encouraging that the implementation of Dirichlet and Neumann conditions in the quantum theory is straightforward, and we thus expect that the boundary condition implementation described in this paper can be used "as is" for the description of strings with either freely propagating endpoints, or with endpoints restricted to lie on D-branes [64, 65]. Finally, a challenging open problem would be to construct a higher-dimensional generalization of this QGPE. This would arise as the TDVP equation for the variational set of continuous projected-entangled pair states [66], which are less well studied and understood.

## Acknowledgements

We acknowledge discussions with C. Lubich and A. Daley. Research supported by the Research Foundation Flanders (JH), the EPSRC under grant numbers EP/L001578/1 and EP/I031014/1 (VS), the Austrian FWF SFB grants FoQuS and ViCoM, the cluster of excellence EXC 201 "Quantum Engineering and Space-Time Research" and the European grants SISQ, QUTE and QFTCMPS.

## A  Derivation of the quantum Gross-Pitaevskii equation and the quantum Bogoliubov-de Gennes equations

### A.1  Quantum Gross-Pitaevskii equation

We start from the fully generic cMPS definition

$$|\Psi[Q,R,\boldsymbol{v}_1,\boldsymbol{v}_2]\rangle = \boldsymbol{v}_1^\dagger \mathcal{P}e^{\int_{x_1}^{x_2} Q(x)\otimes\hat{1}+R(x)\otimes\hat{\psi}^\dagger(x)\,\mathrm{d}x}\boldsymbol{v}_2\,|\Omega\rangle, \tag{16}$$

with virtual dimension $D$, and the Lieb-Liniger Hamiltonian

$$\hat{H} = \int_{x_1}^{x_2}\mathrm{d}x\,\frac{\mathrm{d}\hat{\psi}^\dagger}{\mathrm{d}x}(x)\frac{\mathrm{d}\hat{\psi}}{\mathrm{d}x}(x) + v(x)\hat{\psi}^\dagger(x)\hat{\psi}(x) + g\hat{\psi}^\dagger(x)\hat{\psi}^\dagger(x)\hat{\psi}(x)\hat{\psi}(x). \tag{17}$$

The energy expectation value was computed in Refs. [38,40] and is given by

$$\langle\Psi|\hat{H}|\Psi\rangle = \int\mathrm{d}x\,\langle\rho_L(x)|\mathcal{D}_xR(x)\otimes\overline{\mathcal{D}_xR(x)} + v(x)R(x)\otimes\overline{R(x)} + gR(x)^2\otimes\overline{R(x)}^2|\rho_R(x)\rangle \tag{18}$$

with the left and right density matrices $\rho_L(x)$ and $\rho_R(x)$ defined by

$$\rho_L(x_1) = \boldsymbol{v}_1\boldsymbol{v}_1^\dagger, \qquad \frac{\mathrm{d}\rho_L}{\mathrm{d}x}(x) = Q(x)^\dagger\rho_L(x) + \rho_L(x)Q(x) + R(x)^\dagger\rho_L(x)R(x), \tag{19a}$$

$$\rho_R(x_2) = \boldsymbol{v}_2\boldsymbol{v}_2^\dagger, \qquad \frac{\mathrm{d}\rho_R}{\mathrm{d}x}(x) = -\big[Q(x)\rho_R(x) + \rho_R(x)Q(x)^\dagger + R(x)\rho_R(x)R(x)^\dagger\big] \tag{19b}$$

and with

$$\mathcal{D}_xR(x) = \frac{\mathrm{d}R}{\mathrm{d}x}(x) + [Q(x),R(x)] \tag{20}$$

the spatial covariant derivative of R(x).

To facilitate the rest of the derivation, we also introduce the notation $\hat{M}(y,z) = \mathcal{P}e^{\int_y^z \mathrm{d}x\,Q(x)\otimes\hat{1}+R(x)\otimes\hat{\psi}^\dagger(x)}$. A general tangent vector is obtained by computing the variation in the state $|\Psi[Q,R,\boldsymbol{v}_1,\boldsymbol{v}_2]\rangle$ under a generic variation $Q(x)\to Q(x)+\delta Q(x)$, $R(x)\to R(x)+\delta R(x)$, $\boldsymbol{v}_{1,2}\to \boldsymbol{v}_{1,2}+\delta\boldsymbol{v}_{1,2}$. The result is denoted as the state $|\Phi\rangle$ given by

$$|\Phi[\delta Q,\delta R,\delta\boldsymbol{v}_1,\delta\boldsymbol{v}_2]\rangle = \int_{x_1}^{x_2}\boldsymbol{v}_1^\dagger\hat{M}(x_1,x)\big[\delta Q(x)\otimes\hat{1} + \delta R(x)\otimes\hat{\psi}^\dagger(x)\big]\hat{M}(x,x_2)\,|\Omega\rangle\,\mathrm{d}x$$
$$+ \delta\boldsymbol{v}_1^\dagger\hat{M}(x_1,x_2)\boldsymbol{v}_2\,|\Omega\rangle + \boldsymbol{v}_1^\dagger\hat{M}(x_1,x_2)\delta\boldsymbol{v}_2\,|\Omega\rangle. \tag{21}$$

To properly deal with the boundary conditions, it is useful to derive the QGPE following the general recipe of the TDVP, i.e. as the Euler-Lagrange equations corresponding to extremizing the classical action

$$S[Q,R,\boldsymbol{v}_1,\boldsymbol{v}_2] = \int\mathrm{d}t\int_{x_1}^{x_2}\mathrm{d}x\,\langle\Psi[\overline{Q},\overline{R},\overline{\boldsymbol{v}}_1,\overline{\boldsymbol{v}}_2]|i\frac{\mathrm{d}}{\mathrm{d}t} - \hat{H}|\Psi[Q,R,\boldsymbol{v}_1,\boldsymbol{v}_2]\rangle \tag{22}$$

We can easily derive the Euler-Lagrange equations by considering variations with respect to the complex conjugates of the variational parameters, which are treated as independent and appear only in the bra. Using our definition of tangent vectors, this immediately leads to the condition (using dots for time derivatives)

$$i \langle \Phi[\overline{\delta Q}, \overline{\delta R}, \overline{\delta \nu_1}, \overline{\delta \nu_2}] | \Phi[\dot{Q}, \dot{R}, \dot{\nu}_1, \dot{\nu}_2] \rangle = \langle \Phi[\overline{\delta Q}, \overline{\delta R}, \overline{\delta \nu_1}, \overline{\delta \nu_2}] | \hat{H} | \Psi[Q, R, \nu_1, \nu_2] \rangle \qquad (23)$$

for any possible variation. This is indeed equivalent to the geometric formulation of the TDVP in Eq. (3) of the main text. More generally, it tells us that we can find the tangent space projection $|\Phi[V, W, \boldsymbol{w}_1, \boldsymbol{w}_2]\rangle = \hat{P}_\Psi |\Theta\rangle$ of an arbitrary state $|\Theta\rangle$ —not necessarily $\hat{H}|\Psi[Q, R, \nu_1, \nu_2]\rangle$— by choosing $V(x)$, $W(x)$ and $\boldsymbol{w}_{1,2}$ such that

$$\langle \Phi[\overline{V}', \overline{W}', \overline{\boldsymbol{w}}_1', \overline{\boldsymbol{w}}_2'] | \Phi[V, W, \boldsymbol{w}_1, \boldsymbol{w}_2] \rangle = \langle \Phi[\overline{V}', \overline{W}', \overline{\boldsymbol{w}}_1', \overline{\boldsymbol{w}}_2'] | \Theta \rangle \qquad (24)$$

for all possible $\overline{V}'(x) = \overline{\delta Q}(x)$, $\overline{W}'(x) = \overline{\delta R}(x)$ and $\overline{\boldsymbol{w}}_{1,2}' = \overline{\delta \nu}_{1,2}$. The left hand side contains the overlap of two different tangent vectors and is given by

$$\langle \Phi[V', W', \boldsymbol{w}_1', \boldsymbol{w}_2'] | \Phi[V, W, \boldsymbol{w}_1, \boldsymbol{w}_2] \rangle = \int_{x_1}^{x_2} dx \, \langle \rho_L(x) | W(x) \otimes \overline{W}'(x) | \rho_R(x) \rangle$$

$$+ \int_{x_1}^{x_2} dx \int_x^{x_2} dy \, \langle \rho_L(x) | \big( V(x) \otimes \mathbb{1} + W(x) \otimes \overline{R}(x) \big) E(x, y) \big( \mathbb{1} \otimes \overline{V}'(y) + R(y) \otimes \overline{W}'(y) \big) | \rho_R(y) \rangle$$

$$+ \int_{x_1}^{x_2} dx \int_{x_1}^x dy \, \langle \rho_L(y) | \big( \mathbb{1} \otimes \overline{V}'(y) + R(y) \otimes \overline{W}'(y) \big) E(y, x) \big( V(x) \otimes \mathbb{1} + W(x) \otimes \overline{R}(x) \big) | \rho_R(x) \rangle$$

$$+ \int_{x_1}^{x_2} dx [\boldsymbol{\nu}_1^\dagger \otimes \overline{\boldsymbol{w}}_1'^\dagger] E(x_1, x) \big( V(x) \otimes \mathbb{1} + W(x) \otimes \overline{R}(x) \big) | \rho_R(x) \rangle$$

$$+ \int_{x_1}^{x_2} dx \, \langle \rho_L(x) | \big( V(x) \otimes \mathbb{1} + W(x) \otimes \overline{R}(x) \big) E(x, x_2) [\boldsymbol{\nu}_2 \otimes \overline{\boldsymbol{w}}_2']$$

$$+ \int_{x_1}^{x_2} dy \, [\boldsymbol{w}_1^\dagger \otimes \overline{\boldsymbol{\nu}}_1^\dagger] E(x_1, y) \big( \mathbb{1} \otimes \overline{V}'(y) + R(y) \otimes \overline{W}'(y) \big) | \rho_R(x) \rangle + [\boldsymbol{w}_1^\dagger \otimes \overline{\boldsymbol{w}}_1'^\dagger] | \rho_R(x_1) \rangle$$

$$+ [\boldsymbol{w}_1^\dagger \otimes \overline{\boldsymbol{\nu}}_1^\dagger] E(x_1, x_2) [\boldsymbol{\nu}_2 \otimes \overline{\boldsymbol{w}}_2']$$

$$+ \int_{x_1}^{x_2} dy \, \langle \rho_L(y) | \big( \mathbb{1} \otimes \overline{V}'(y) + R(y) \otimes \overline{W}'(y) \big) [\boldsymbol{w}_2 \otimes \overline{\boldsymbol{\nu}}_2] + \langle \rho_L(x_2) | [\boldsymbol{w}_2 \otimes \overline{\boldsymbol{w}}_2']$$

$$+ [\boldsymbol{\nu}_1^\dagger \otimes \overline{\boldsymbol{w}}_1'^\dagger] E(x_1, x_2) [\boldsymbol{w}_2 \otimes \overline{\boldsymbol{\nu}}_2],$$

where the terms on the first 5 lines corresponds to all contributions of non-zero $V$ and $W$, and the terms on lines 6 and 7 correspond to non-zero $\boldsymbol{w}_1$ and non-zero $\boldsymbol{w}_2$, respectively. Here, we have introduced a new notation

$$E(x, y) = \mathcal{P} \exp \left( \int_x^y Q(z) \otimes \mathbb{1} + \mathbb{1} \otimes \overline{Q}(z) + R(z) \otimes \overline{R}(z) \, dz \right). \qquad (25)$$

It is the continuum equivalent of the (product of) MPS transfer matrices and allows to *e.g.* write

$$\langle \rho_L(x) | = [\boldsymbol{\nu}_1^\dagger \otimes \overline{\boldsymbol{\nu}}_1^\dagger] E(x_1, x), \qquad \qquad | \rho_R(x) \rangle = E(x, x_2) [\boldsymbol{\nu}_2 \otimes \overline{\boldsymbol{\nu}}_2]. \qquad (26)$$

Let us now first compute $\hat{H} | \Psi[Q, R, \nu_1, \nu_2] \rangle$ itself. Using the rules from Ref. [40] and by

applying partial integration to the kinetic energy term, we obtain

$$
\hat{H}\,|\Psi[Q,R,\boldsymbol{v}_1,\boldsymbol{v}_2]\rangle = \int_{x_1}^{x_2} \boldsymbol{v}_1^\dagger \hat{M}(x_1,x)\big(-\mathcal{D}_x^2 R(x)+v(x)R(x)\big)\otimes\hat{\psi}^\dagger(x)\hat{M}(x,x_2)\boldsymbol{v}_2\,|\Omega\rangle\,\mathrm{d}x
$$
$$
+\int_{x_1}^{x_2}\boldsymbol{v}_1^\dagger \hat{M}(x_1,x)\big(gR(x)^2-[R(x),\mathcal{D}_x R(x)]\big)\otimes\big(\hat{\psi}^\dagger(x)\big)^2\hat{M}(x,x_2)\boldsymbol{v}_2\,|\Omega\rangle\,\mathrm{d}x
$$
$$
-\boldsymbol{v}_1^\dagger\mathcal{D}_x R(x_1)\otimes\psi^\dagger(x_1)\hat{M}(x_1,x_2)\boldsymbol{v}_2\,|\Omega\rangle+\boldsymbol{v}_1^\dagger\hat{M}(x_1,x_2)\mathcal{D}_x R(x_2)\otimes\psi^\dagger(x_2)\boldsymbol{v}_2\,|\Omega\rangle. \quad (27)
$$

Given the linearity of the tangent space projector, we can compute the projection of the 4 different terms separately and add the result:

1. The first term of Eq. (27) is already in the explicit form of a tangent vector $|\Phi[V,W,\boldsymbol{w}_1,\boldsymbol{w}_2]\rangle$ with $W(x)=-\mathcal{D}_x^2 R(x)+v(x)R(x)$ and $V(x)=0$, $\boldsymbol{w}_{1,2}=0$. It does not need to be projected.

2. The second term is of the form $|\Theta\rangle=\int_{x_1}^{x_2}\boldsymbol{v}_1^\dagger\hat{M}(x_1,x)B(x)\otimes\big(\hat{\psi}^\dagger(x)\big)^2\hat{M}(x,x_2)\boldsymbol{v}_2\,|\Omega\rangle\,\mathrm{d}x$, where $B(x)=gR(x)^2-[R(x),\mathcal{D}_x R(x)]$. We obtain

$$
\langle\Phi[V',W',\boldsymbol{w}_1',\boldsymbol{w}_2']|\Theta\rangle=\int_{x_1}^{x_2}\mathrm{d}x\,\langle\rho_L(x)|B(x)\otimes\big(\overline{R}(x)\overline{W}'(x)+\overline{W}'(x)\overline{R}(x)\big)|\rho_R(x)\rangle
$$
$$
+\int_{x_1}^{x_2}\mathrm{d}x\int_{x}^{x_2}\mathrm{d}y\,\langle\rho_L(x)|\big(B(x)\otimes\overline{R}(x)^2\big)E(x,y)\big(\mathbb{1}\otimes\overline{V}'(y)+R(y)\otimes\overline{W}'(y)\big)|\rho_R(y)\rangle
$$
$$
+\int_{x_1}^{x_2}\mathrm{d}x\int_{x_1}^{x}\mathrm{d}y\,\langle\rho_L(y)|\big(\mathbb{1}\otimes\overline{V}'(y)+R(y)\otimes\overline{W}'(y)\big)E(y,x)\big(B(x)\otimes\overline{R}(x)^2\big)|\rho_R(x)\rangle
$$
$$
+\int_{x_1}^{x_2}\mathrm{d}x[\boldsymbol{v}_1^\dagger\otimes\overline{\boldsymbol{w}}_1'^{\dagger}]E(x_1,x)\big(B(x)\otimes\overline{R}(x)^2\big)|\rho_R(x)\rangle
$$
$$
+\int_{x_1}^{x_2}\mathrm{d}x\,\langle\rho_L(x)|\big(B(x)\otimes\overline{R}(x)^2\big)E(x,x_2)[\boldsymbol{v}_2\otimes\overline{\boldsymbol{w}}_2']
$$

One can easily verify that by choosing

$$
V(x)=-\rho_L(x)^{-1}R(x)^\dagger\rho_L(x)B(x)\rho_R(x)R(x)^\dagger\rho_R(x)^{-1}
$$
$$
W(x)=+\rho_L(x)^{-1}R(x)^\dagger\rho_L(x)B(x)+B(x)\rho_R(x)R(x)^\dagger\rho_R(x)^{-1}
$$
$$
\boldsymbol{w}_1^\dagger=0
$$
$$
\boldsymbol{w}_2=0
$$

every single line of $\langle\Phi[V',W',\boldsymbol{w}_1',\boldsymbol{w}_2']|\Theta\rangle$ matches with the corresponding line in the first lines of $\langle\Phi[V',W',\boldsymbol{w}_1',\boldsymbol{w}_2']|\Phi[V,W,\boldsymbol{w}_1,\boldsymbol{w}_2]\rangle$, whereas the last two lines of the latter vanish because of the choice of $\boldsymbol{w}_{1,2}=0$.

3. Next we deal with the third term $|\Theta\rangle=-\boldsymbol{v}_1^\dagger\mathcal{D}_x R(x_1)\otimes\psi^\dagger(x_1)\hat{M}(x_1,x_2)\boldsymbol{v}_2\,|\Omega\rangle$, resulting in

$$
\langle\Phi[V',W',\boldsymbol{w}_1',\boldsymbol{w}_2']|\Theta\rangle=-[\boldsymbol{v}_1^\dagger\otimes\overline{\boldsymbol{v}}_1^\dagger]\mathcal{D}_x R(x_1)\otimes\overline{W}'(x_1)|\rho_R(x_1)\rangle
$$
$$
-\int_{x_1}^{x_2}\mathrm{d}y[\boldsymbol{v}_1^\dagger\otimes\overline{\boldsymbol{v}}_1^\dagger]\mathcal{D}_x R(x_1)\otimes\overline{R}(x_1)E(x_1,y)\big(\mathbb{1}\otimes\overline{V}'(y)+R(y)\otimes\overline{W}'(y)\big)|\rho_R(y)\rangle
$$
$$
-[\boldsymbol{v}_1^\dagger\otimes\overline{\boldsymbol{w}}_1'^{\dagger}]\mathcal{D}_x R(x_1)\otimes\overline{R}(x_1)|\rho_R(x_1)\rangle
$$
$$
-[\boldsymbol{v}_1^\dagger\otimes\overline{\boldsymbol{v}}_1^\dagger]\mathcal{D}_x R(x_1)\otimes\overline{R}(x_1)E(x_1,x_2)[\boldsymbol{v}_2\otimes\overline{\boldsymbol{w}}_2']
$$

Now we have to consider the effect of the boundary conditions. If $R(x_1)$ is fixed as $R(x_1) = a\mathbb{1}$ (Dirichlet condition), then the corresponding variation $\overline{W}'(x_1) = 0$ such that the first term vanishes and the remaining terms match the sixth line of $\langle\Phi[V', W', \boldsymbol{w}_1', \boldsymbol{w}_2']|\Phi[V, W, \boldsymbol{w}_1, \boldsymbol{w}_2]\rangle$ by choosing

$$V(x) = 0, \qquad W(x) = 0, \qquad \boldsymbol{w}_1^\dagger = -\overline{a}\boldsymbol{v}_1^\dagger \mathcal{D}_x R(x_1), \qquad \boldsymbol{w}_2 = 0.$$

Indeed, this boundary condition corresponds to fixing the value of the field operator $\hat{\psi}(x_1) = a$, so that $|\Theta\rangle$ can explicitly be rewritten as $|\Theta\rangle = -\overline{a}\boldsymbol{v}_1^\dagger \mathcal{D}_x R(x_1)\hat{M}(x_1, x_2)\boldsymbol{v}_2|\Omega\rangle$, which exactly equals the tangent vector $|\Phi[V, W, \boldsymbol{w}_1, \boldsymbol{w}_2]\rangle$ for this choice of the parameters.

Alternatively, if we do not fix $R(x_1)$, then the variation $\overline{W}'(x_1)$ automatically enforces the Neumann conditions $\boldsymbol{v}_1^\dagger \mathcal{D}_x R(x_1) = 0$ at any point in time, provided that $\rho_R(x_1)$ is full rank. Under this condition, we also obtain $|\Theta\rangle = 0$, which corresponds to inserting $\mathcal{D}_x R(x_1) = 0$ in the parameters above.

4. The last term of Eq. (27) can be dealt with similarly and is an explicit tangent vector corresponding to the choice

$$V(x) = 0, \qquad W(x) = 0, \qquad \boldsymbol{w}_1^\dagger = 0, \qquad \boldsymbol{w}_2 = \overline{b}\mathcal{D}_x R(x_2)\boldsymbol{v}_2.$$

in case of the Dirichlet condition $R(x_2) = b\mathbb{1}$. In the case of the Neumann condition, it is also zero.

Hence, for the boundary conditions here considered, only the second term of Eq. (27) needed to be projected onto the tangent space and, when considering the full Schrödinger equation, would be responsible for taking the exact evolution out of the manifold.

Grouping everything together gives rise to $|\Phi[V, W, \boldsymbol{w}_1, \boldsymbol{w}_2]\rangle = \hat{P}_\Psi \hat{H} |\Psi[Q, R, \boldsymbol{v}_1, \boldsymbol{v}_2]\rangle$ with

$$
\begin{aligned}
V(x) &= -\rho_L(x)^{-1}R(x)^\dagger \rho_L(x)\big(gR(x)^2 - [R(x), \mathcal{D}_x R(x)]\big)\rho_R(x)R(x)^\dagger \rho_R(x)^{-1} \\
W(x) &= -\mathcal{D}_x^2 R(x) + v(x)R(x) + \rho_L(x)^{-1}R(x)^\dagger \rho_L(x)\big(gR(x)^2 - [R(x), \mathcal{D}_x R(x)]\big) \\
&\quad + \big(gR(x)^2 - [R(x), \mathcal{D}_x R(x)]\big)\rho_R(x)R(x)^\dagger \rho_R(x)^{-1} \\
\boldsymbol{w}_1^\dagger &= -\overline{a}\boldsymbol{v}_1^\dagger \mathcal{D}_x R(x_1) \\
\boldsymbol{w}_2 &= +\overline{b}\mathcal{D}_x R(x_2)\boldsymbol{v}_2 \,,
\end{aligned}
$$

which we have to equate to $i|\Phi[\dot{Q}, \dot{R}, \dot{\boldsymbol{v}}_1, \dot{\boldsymbol{v}}_2]\rangle$. Matching the parameters as $\dot{Q} = V$ etc gives rise to the gauge covariatn QGPE of the main text for the specific choice of $P = 0$, though this choice is not unique. Indeed, one can note that the physical tangent vector $|\Phi[V, W, \boldsymbol{w}_1, \boldsymbol{w}_2]\rangle$ does not change under a substitution

$$
\begin{aligned}
V(x) \to V(x) + [P(x), \quad Q(x)], W(x) &\to W(x) + [P(x), R(x)], \\
\boldsymbol{w}_1^\dagger \to \boldsymbol{w}_1^\dagger - \boldsymbol{v}_1^\dagger P(x_1) \quad, \boldsymbol{w}_2 &\to \boldsymbol{w}_2 + P(x_2)\boldsymbol{v}_2,
\end{aligned}
\tag{28}
$$

which is how the fully gauge covariant formulation of the QGPE is obtained.

## A.2   Derivation of the quantum Bogoliubov-de Gennes Equations

A stationary solution $Q_0$ and $R_0$ of the QGPE parameterizes a variationally optimal cMPS ground state approximation for a given Hamiltonian $\hat{H}_0$. Upon applying a perturbation $\hat{H}_1$ (possibly time-dependent), we can expand the QGPE to first order around $Q_0$ and $R_0$. In the following, we set $\hat{H}_0$ equal to the translation invariant Lieb-Liniger Hamiltonian ($v(x) = v_0 = -\mu$

with $\mu$ the chemical potential) in the thermodynamic limit, so that the stationary solution correspond to $x$ (and $t$) independent matrices $Q_0$ and $R_0$, which we assume to be in the left-canonical form, i.e. $Q_0 + Q_0^\dagger + R_0^\dagger R_0 = 0$. Associated with this solution is a right density matrix $\rho_{R,0}$ satisfying

$$Q_0 \rho_{R,0} + \rho_{R,0} Q_0^\dagger + R_0 \rho_{R,0} R_0^\dagger = 0$$

and a matrix $P_0 = -iR_0^\dagger[Q_0, R_0] + iF_0$ where $F_0$ is the solution of the linear system

$$-Q_0^\dagger F_0 - F_0 Q_0 - R_0^\dagger F_0 R_0 = [Q_0, R_0]^\dagger [Q_0, R_0] - \mu R_0^\dagger R_0 + g(R_0^\dagger)^2 R_0^2.$$

We now consider a perturbation given by $\hat{H}_1(t) = \epsilon \int dx\, \tilde{v}(x,t)\hat{\psi}^\dagger(x)\hat{\psi}(x)$. The ansatz for the new solution of the QGPE is then given by

$$R(x,t) = R_0 + \epsilon \tilde{R}(x,t), \qquad Q(x,t) = Q_0 + \epsilon \tilde{Q}(x,t),$$
$$\rho_R(x,t) = \rho_{R,0} + \epsilon \tilde{\rho}_R(x,t), \qquad \rho_L(x,t) = \mathbb{1} + \epsilon \tilde{\rho}_L(x,t), \tag{29}$$

where $\rho_{L,R}(x,t)$ are the left and right density matrices. By expanding the relevant equations to first order in $\epsilon$, we obtain

$$\partial_x \tilde{\rho}_L(x,t) - Q_0^\dagger \tilde{\rho}_L(x,t) + \tilde{\rho}_L(x,t)Q_0 + R_0^\dagger \tilde{\rho}_L(x,t)R_0 =$$
$$\tilde{Q}(x,t) + \tilde{Q}^\dagger(x,t) + R_0^\dagger \tilde{R}(x,t) + \tilde{R}^\dagger(x,t)R_0 \tag{30}$$

$$\partial_x \tilde{\rho}_R(x,t) + Q_0 \tilde{\rho}_R(x,t) + \tilde{\rho}_R(x,t)Q_0^\dagger + R_0 \tilde{\rho}_R(x,t)R_0^\dagger =$$
$$-\big[\tilde{Q}(x,t)\rho_{R,0} + \rho_{R,0}\tilde{Q}^\dagger(x,t) + \tilde{R}(x,t)\rho_{R,0}R_0^\dagger + R_0 \rho_{R,0}\tilde{R}^\dagger(x,t)\big]. \tag{31}$$

We furthermore choose

$$P(x,t) = P_0 + \epsilon \tilde{P}(x,t) = P_0 + \epsilon\big(-i\tilde{R}^\dagger[Q_0, R_0] - iR_0^\dagger \tilde{R}_{\rm kin}(x,t) + i\tilde{F}(x,t)\big)$$

with $\tilde{R}_{\rm kin}(x,t) = [\tilde{Q}(x), R_0] + [Q_0, \tilde{R}(x)] + \partial_x \tilde{R}(x)$ and $\tilde{F}(x,t)$ the solution of

$$\partial_x \tilde{F} = \tilde{F}Q_0 + F_0\tilde{Q} + \tilde{Q}^\dagger F_0 + Q_0^\dagger \tilde{F} + \tilde{R}^\dagger F_0 R_0 + R_0^\dagger \tilde{F}R_0 + R_0^\dagger F_0 \tilde{R} + [Q_0, R_0]^\dagger \tilde{R}_{\rm kin} + \tilde{R}_{\rm kin}^\dagger[Q_0, R_0]$$
$$+ g\big(\tilde{R}^\dagger R_0^\dagger R_0 R_0 + R_0^\dagger \tilde{R}^\dagger R_0 R_0 + R_0^\dagger R_0^\dagger \tilde{R} R_0 + R_0^\dagger R_0^\dagger R_0 \tilde{R}\big) + v_0 \tilde{R}^\dagger R_0 + v_0 R_0^\dagger \tilde{R} + \tilde{v}R_0^\dagger R_0, \tag{32}$$

where we henceforth omit the $(x,t)$ dependence of the $\tilde{\ }$ quantities. With this choice, we are assured that $\partial_t \tilde{Q} + R_0^\dagger \partial_t \tilde{R} = 0$ and, integrating this from the initial time when $\tilde{Q} = \tilde{R} = 0$, we obtain $\tilde{Q} + R_0^\dagger \tilde{R} = 0$. This equality assures that $Q(x)$ and $R(x)$ form a left canonical representation to first order in $\epsilon$. In particular, this makes the right hand side of Eq. (30) equal to zero, so that the solution of that equation is given by $\tilde{\rho}_L = 0$.

The linearized QGPE itself is then given by

$$i\partial_t \tilde{R}\rho_{R,0} = -\partial_x^2 \tilde{R}\rho_{R,0} - 2[Q_0, \partial_x \tilde{R}]\rho_{R,0} - [\partial_x \tilde{Q}, R_0]\rho_{R,0}$$
$$- [\tilde{Q}, [Q_0, R_0]]\rho_{R,0} - [Q_0, [\tilde{Q}, R_0]]\rho_{R,0} - [Q_0, [Q_0, \tilde{R}]]\rho_{R,0} - [Q_0, [Q_0, R_0]]\tilde{\rho}_R$$
$$+ g\big((\tilde{R}^\dagger R_0^2 + R_0^\dagger \tilde{R}R_0 + R_0^\dagger R_0 \tilde{R})\rho_{R,0} + R_0^\dagger R_0^2 \tilde{\rho}_R + \tilde{R}R_0 \rho_{R,0}R_0^\dagger + R_0 \tilde{R}\rho_{R,0}R_0^\dagger + R_0^2 \tilde{\rho}_R R_0^\dagger + R_0^2 \rho_{R,0}\tilde{R}^\dagger\big)$$
$$+ v_0 \tilde{R}\rho_{R,0} + v_0 R_0 \tilde{\rho}_R + \tilde{v}R_0 \rho_{R,0}$$
$$+ [\tilde{R}_{\rm kin}, R_0]\rho_{R,0}R_0^\dagger + [[Q_0, R_0], \tilde{R}]\rho_{R,0}R_0^\dagger + [[Q_0, R_0], R_0]\tilde{\rho}_R R_0^\dagger + [[Q_0, R_0], R_0]\rho_{R,0}\tilde{R}^\dagger$$
$$+ [\tilde{R}, R_0^\dagger][Q_0, R_0]\rho_{R,0} + [R_0, \tilde{R}^\dagger][Q_0, R_0]\rho_{R,0} + [R_0, R_0^\dagger]\tilde{R}_{\rm kin}\rho_{R,0} + [R_0, R_0^\dagger][Q_0, R_0]\tilde{\rho}_R$$
$$+ [\tilde{F}, R_0]\rho_{R,0} + [F_0, \tilde{R}]\rho_{R,0} + [F_0, R_0]\tilde{\rho}_R \tag{33}$$

where we can further substitute $\tilde{Q} = -R_0^\dagger \tilde{R}$. Equations (31), (32) and (33) form a set of linear partial differential equations with as only source term the perturbed potential $\tilde{v}(x,t)$ and with mixing between $\tilde{R}$ and $\tilde{R}^\dagger$. Therefore, if $\tilde{v}(x,t) = \cos(kx - \omega t)$ for certain $k$ and $\omega$, we can make the ansatz

$$\tilde{R}(x) = e^{\mathrm{i}(kx-\omega t)} R_+ + e^{-\mathrm{i}(kx-\omega t)} R_-, \tilde{F}(x) = e^{\mathrm{i}(kx-\omega t)} F_+ + e^{-\mathrm{i}(kx-\omega t)} F_-,$$

$$\tilde{\rho}_{\mathrm{R}}(x) = e^{\mathrm{i}(kx-\omega t)} \rho_+ + e^{-\mathrm{i}(kx-\omega t)} \rho_-.$$

Note that both $\tilde{F}(x)$ and $\tilde{\rho}_{\mathrm{R}}(x)$ are hermitian matrices which means that $F_-^\dagger = F_+$ and $\rho_-^\dagger = \rho_+$. Inserting this ansatz into the relevant equations allows to express everything in terms of $R_\pm$ and finally gives rise to the quantum Bogoliubov-de Gennes equation [Eq. (10) in the main text]. This equation can be iteratively solved at a computational cost of $\mathcal{O}(D^3)$. The same approach still works if $\tilde{v}(x,t)$ contains $N$ different Fourier modes, where the complexity will increase linearly with $N$.

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
