# Peer review of "Quantum Gross-Pitaevskii Equation"

_SciPost Physics, doi:SciPost Phys. 3, 006 (2017)_

## Round 3 · Referee Report · Anonymous · 2016-12-21

Strengths

1) Interesting result of possibly broad relevance, extending conventional (mean-field) method to deal with quantum effects.
2) Concisely and well written.

Weaknesses

1) Some minor details need to be added (see changes).
2) Perhaps need to emphasize shorter-term applications of result (i.e. QBdG equations) given current challenges in numerical implementation of the QGPE.

Report

The authors develop an interesting extension of the well-known Gross-Pitaevskii equation (GPE) within the context of 1D systems, the Quantum Gross-Pitaevskii equation (QGPE). This work is obviously of interest, given that whilst the GPE works excellently in describing many features of ultracold gases in 3D (where bosonic gases may form a BEC with long-range order), it is known to be invalid for 1D systems where in general there is a lack of long-range order (e.g., quasi-condensation of Bose gases in 1D) and the role of quantum correlations can be important (hence the breakdown of mean-field methods). To overcome these issues the authors develop the QGPE by a combination of the usual Dirac-Frenkel time-dependent variational principle with a cMPS (continuous matrix product state) ansatz for the wavefunction (which reduces to the usual mean-field variational ansatz under appropriate limits). Using the cMPS representation allows the QGPE to capture quantum correlations above the mean-field level, which are crucial for the description of 1D systems.

A large portion of the manuscript is devoted to the mathematical and technical basis of the QGPE, and further detail is appropriately given in the supplementary material. They also discuss that, importantly, the actual numerical implementation of the QGPE may be a non-trivial though not intractable problem. This is mainly due to the treatment of spatial derivatives (which occur within the nonlinear terms of the QGPE equation) and the appearance of low-rank matrices chracterising the boundaries of the system. Whilst not being specific, the authors do not claim that any of these problems are insurmountable in principle.

Given the current limitations of the QGPE, the authors discuss a useful implementation of the QGPE to derive a set of effective 'Quantum Bogoliubov-de Gennes equations' (QBdG). Whilst this section is concisely written, they demonstrate that the QBdG equations can be used to capture (known) physics beyond the scope of 1D mean-field methods. Specifically, they examine the response of a 1D system to a driven external potential. I would comment that, given this is the only illustrative application of the QGPE given in the manuscript (c/o the above discussed limitations of the full QGPE), that perhaps it be more emphasized within the introduction or abstract of the paper.

Lastly, the authors give a general outlook for the QGPE and possible avenues of utility. Here, I wonder if the authors could perhaps comment/speculate further on the - in principle - feasibility of the QGPE with respect to other known methods for treating the full quantum problem (in 1D). Specifically, the authors note that it would be interesting to compare the predictions of the QGPE to Ref. [59], which implements phase-space methods. Given that, in general, the various phase-space methods suffer from a range of `understood' problems such as truncation error and stability issues (beyond short-times), it would be interesting for the authors to comment whether they believe the QGPE would perhaps be a more valid/useful approach in some instances.

Overall, I believe the manuscript is well-written and presents a interesting result. I recommend publication with changes as detailed.

Requested changes

1) Mention that one can derive effective QBdG equations c/o the QGPE earlier in the manuscript, i.e. introduction.
2) Comment further on whether the QGPE could be a more effective approach for some systems in 1D than known methods (see above comment in report).
2) Fig. 1: \alpha is not defined explicitly in the text or caption. This should be done, particularly as currently it is effectively defined by a label in a subsequent plot.
3) Fig. 1: Similarly, \delta<\rho(x)> (= <\rho(x)> - \rho) is not explicitly defined in text or caption.
4) Fig. 1: There is a horizontal line in the lower plot of Fig. 1 which is never defined. I suspect this is the period of the density fluctuations. Either this should be commented on in the caption, labelled appropriately or removed.

  • validity: top
  • significance: high
  • originality: good
  • clarity: high
  • formatting: excellent
  • grammar: excellent

Author Jutho Haegeman on 2017-03-09
(in reply to Report 1 on 2016-12-21)

We thank the referee for his/her careful reading and positive evaluation of our manuscript. We agree with all comments and requested changes and have incorporated them in the new submission. In particular, we now discuss the quantum Bogoliubov de Gennes equations (and its application) in the abstract and introduction. Furthermore, we have added a sentence to clarify what we expect from a comparison of cMPS results with results from phase space methods for dynamical simulations. Finally, we have adapted Figure 1 and the surrounding discussion in order to properly define all quantities, according to the suggestions of the referee.

---

## Round 3 · Referee Report · Anonymous · 2017-2-1

Strengths

New results relevant for many particle physics of interacting 1d systems

Weaknesses

Not enough evidence of the validity of the approach

Very difficult to read, can only be understood by the community dealing with matrix product states.

Length is not appropriate (too short)

Report

The manuscript proposes a novel ansatz for mean-field approach to one
dimensional interacting bosons. The ansatz replaces a scalar Gross-Pitaevskii
wavefunction by a D-dimensional matrix-valued order parameter. This approach
is claimed to be reliable for treating strongly interacting regime of
Lieb-Liniger model. While I do not object the validity of this approach I feel
that its verification is somewhat insufficient. The provided calculation of
the density-density response function is not well explained and there is no
comparison with the results based on the exact solution. There is plenty of
such results in the literature either based on Luttinger liquid approach (for
all valued of interactions) or the mapping to free fermions (in the strongly
interacting regime). The authors should also explain how the dimensionality D
of the matrix-valued order parameter scales with the interactions and what is
the underlying physics.

In addition the paper is quite obscure and very difficult to read. The length
seems to be to short. I suggest incorporating the Supplementary material into
the main text and making the paper more self-contained otherwise it will only
be understood by specialists in the area of matrix product states. I recommend
major revision or resubmission to a more specialized journal.

Requested changes

See report above

  • validity: good
  • significance: good
  • originality: high
  • clarity: low
  • formatting: reasonable
  • grammar: reasonable

Author Jutho Haegeman on 2017-03-09
(in reply to Report 2 on 2017-02-01)

We thank the referee for reading our manuscript. Unfortunately, we get the impression that the referee has misinterpreted the central goal of this paper, and what we set out to do precisely. Any suggestions that would help to clarify the central goals of our paper are appreciated.

In particular, we do not attempt to provide a new method for specifically dealing with the strongly interacting regime of the Lieb-Liniger model. Rather, the narrative of our manuscript is to explain how, starting from the variational class of wavefunctions known as continuous matrix product states, we obtain a natural generalization of the Gross-Pitaevskii equation by applying the time-dependent variatonal principle. From this so-called quantum Gross-Pitaevskii equation emerges an interesting mathematical structure (such as local gauge invariance) and by linearizing we also obtain a generalization of the Bogoliubov de-Gennes equations. Continuous matrix product states are a versatile ansatz that are applicable to several models, among which the Lieb-Liniger model, both at weak and strong coupling. As such, we have merely chosen this Hamiltonian because that is also the model that gives rise to the normal Gross-Pitaevskii equation by applying mean field theory (corresponding to a field coherent state as variational ansatz). The accuracy of approximating e.g. the Lieb-Liniger ground state as function of the refinement parameter D has been studied in previous papers and is not the central question here.

As such, it is really the structure of the generalized GPE and its linearization which we explore in this manuscript. The example or application of the formalism serves to the illustrate potential usefulness of this framework, but is not the goal in itself. Any suggestions as to how to clarify the main objective of our paper are very welcome, as already stated above.

However, we find it hard to agree with the claim that this paper can only be understood by MPS specialists. We have specifically tried to avoid MPS terminology as much as possible (and to properly define it otherwise). In particular, we introduce continuous MPS not using an MPS continuum limit, but rather as a natural generalization of coherent states, for which the well-known mean field equations (normal Gross-Pitaevskii) is obtained. It would be great if the referee could point us more specifically to which parts of the paper he or she finds incomprehensible and would require further clarification. However, we do believe that the mathematical derivations ---which again resemble more closely coherent state calculations than typical MPS manipulations--- are rightfully placed in the appendix / supplementary material, and that including these in the main text will only serve to distract the reader from the central narrative of our manuscript.

---

## Round 4 · Referee Report · Anonymous (Referee 1) · 2017-3-30

Strengths
Weaknesses
Report
With respect to the correspondence of the second referee, I am supportive of the authors response. I believe the paper is suitably written as to outline the method as a generalization of the conventional GPE. Whilst it is true that some of the more technical details will only be fully appreciated by an MPS-focused audience, the authors do do write the manuscript at a sufficient level to allow the reader to appreciate their conclusions. Lastly, I disagree with the referees assertion that the paper is too short. Much of the calculation details are indeed attached in the supplemental material, and I believe this is the appropriate place for them. I believe the paper in it's current form is suitably concise for the subject matter. Inclusion of further technical details would only serve to obfuscate the message of the paper.
Requested changes
N/A

---

## Round 4 · Referee Report · Anonymous (Referee 3) · 2017-4-1

Strengths
2 - Concisely written
Weaknesses
Report
I disagree with the second referee on the issues of “length” and “scope” of the paper. More details from the supplementary material in the main text would only obscure the message and make it even harder to read. Furthermore, the paper clearly avoids standard MPS language and is sufficiently general for readers without a tensor network background.
However, a clear disadvantage of the paper is that it is indeed very technical and some parts are hard to follow. For example, I don’t think that phrases like “… since any complex submanifold of Hilbert space is automatically Kähler …” speak to many people. The paper falls short in addressing readers without the appropriate mathematical background. Unfortunately, the last part of the paper that presents a specific example calculation (and that would be accessible to a broader audience) seems unmotivated.
Ideally, the authors would simply show a comparison to an exact calculation as suggested by the referee 2. Then for a general reader the ideas would not seem “insufficiently verified” as referee 2 remarked. At least, however, the authors should explain the physics of Fig. 1 in more detail. Since the authors write in the abstract that their method includes “entanglement and correlations”, they should also explain how this can be seen from Fig. 1. At the moment they only refer to [45] and write that “This response cannot be explained using standard GPE, as interactions are a key ingredient for exciting Type II excitations.” Why is this peak at $2k_F$? What are these excitations? Why are they not described in the GPE and why are they present in the exact solution?
I recommend publication, if the authors add an explanation of the physics.
Requested changes
1 - Add explanation of the physics of Fig. 1

---

## Round 4 · List of Changes

* We now already discuss the quantum Bogoliubov de Gennes equations (and its application) in the abstract and introduction.
* We have expanded our discussion regarding the comparison of cMPS results with results from phase space methods for dynamical simulations.
* We have adapted Figure 1 and the surrounding discussion.

---

## Round 5 · List of Changes

We have extended the discussion of the periodic potential example. We clearly indicate how our framework improves upon the mean field prediction (Bogoliubov theory), which is also shown in the plot. We also discuss the underlying physics that shows up in the response amplitude, namely the signature of the low-lying excitations in the system, which are known to be Lieb's Type I and Type II excitations. Whereas Bogoliubov theory can only access the first type, our method also captures the effect of the second type, which is important at stronger interactions.

---

## Editorial Decision

published